# Compliance with COVID-19 preventive measures among chronic disease patients in Wolaita and Dawuro zones, Southern Ethiopia: A proportional odds model

**Temesgen Bati Gelgelu**[1]*, **Shemsu Nuriye**[1], **Tesfaye Yitna Chichiabellu**[2], **Amene Abebe Kerbo**[1]

1 School of Public Health, College of Health Sciences and Medicine, Wolaita Sodo University, Wolaita Sodo, Ethiopia, 2 School of Nursing, College of Health Sciences and Medicine, Wolaita Sodo University, Wolaita Sodo, Ethiopia

* temesgenbati@gmail.com

## Abstract

### Introduction

So far, shreds of evidence have shown that COVID-19 related hospitalization, serious outcomes, and mortality were high among individuals with chronic medical conditions. However, strict compliance with basic public health measures such as hand washing with soap, social distancing, and wearing masks has been recommended and proven effective in preventing transmission of the infection. Therefore, this study aimed to determine the level of compliance with COVID-19 preventive measures and identify its predictors among patients with common chronic diseases in public hospitals of Southern Ethiopia by applying the proportional odds model.

### Methods

A facility-based cross-sectional study was employed in public hospitals of Southern Ethiopia between February and March 2021. Using a systematic random sampling technique, 419 patients with common chronic diseases were recruited. Data were collected using an Open Data Kit and then submitted to the online server. The proportional odds model was employed, and the level of significance was declared at a p-value of less than 0.05.

### Results

This study revealed that 55.2% (95%CI: 50.4%-59.9%) of the study participants had low compliance levels with COVID-19 preventive measures. The final proportional odds model identified that perceived susceptibility (AOR: 0.91, 95%CI: 0.84, 0.97), cues to action (AOR: 0.89, 95%CI: 0.85, 0.94), having access to drinking water piped into the dwelling (AOR: 0.52, 95%CI: 0.32, 0.84), having no access to any internet (AOR: 0.62, 95%CI: 0.42, 0.92), having no functional refrigerator (AOR: 2.17, 95%CI: 1.26, 3.74), and having poor

**Data Availability Statement:** The minimal anonymized data set necessary to replicate this

study are within the manuscript and its Supporting Information files.

**Funding:** This study was funded by the Wolaita Sodo University with the wsu 41/19/368 grant numbers. Received by TBG. The funders had no role in study design, data collection and analysis, decision to publish, or preparation of the manuscript.

**Competing interests:** The authors have declared that no competing interests exist.

knowledge (AOR: 1.42, 95%CI: 1.02, 1.98) were the independent predictors of low compliance level with COVID-19 preventive measures.

## Conclusion

In the study area, more than half of the participants had low compliance levels with COVID-19 preventive measures. Thus, the identified factors should be considered when designing, planning, and implementing new interventional strategies, so as to improve the participants' compliance level.

## Introduction

So far, shreds of evidence have shown that COVID-19 related hospitalization, serious outcomes, and mortality were high among individuals with chronic medical conditions [1–3]. Accordingly, hypertension, diabetes, and chronic obstructive pulmonary disease were the most common chronic conditions that have been linked to the poor outcome of COVID-19 disease [4–7]. Correspondingly, evidence from a similar place presented that 7.3% of individuals with diabetes, 6.3% of chronic respiratory disease, and 6% of hypertension have died of COVID-19 disease, while only 0.9% of individuals with no underlying chronic medical conditions have died [8]. Similarly, low socio-economic status has been linked to the severe form of COVID-19 [9, 10]. Studies recently conducted among patients with chronic medical conditions in Ethiopia reported that some of the most frequent chronic conditions were hypertension, diabetes, and chronic respiratory diseases [11–13].

Even though there were arguments in the early stages of COVID-19 with respect to mode of transmission, recently most evidence has agreed that it is mainly transmitted through air in the form of droplets and aerosol particles [14–18]. In the meantime, pieces of evidence have suggested strict compliance with basic public health measures such as staying home when sick, covering mouth and nose with a flexed elbow when coughing and sneezing, washing hands often with soap, water, and cleaning frequently touched surfaces or objects are critical to slow the spread of illnesses [3, 19–21]. It also has been proven effective in preventing human-to-human transmission of COVID-19 infection [22].

On 13 March 2020, the first confirmed COVID-19 case was imported to Ethiopia by a Japanese man that came from Burkina Faso. Since that the government of Ethiopia has put public health measures such as closing schools, and restricting large gatherings including religious and social gatherings [23, 24]. Besides, basic prevention measures such as hand washing, social distancing, and wearing masks were the main topics that the government has communicated to the general public via the different media platforms. Remarkably, these public health interventions have held promise to slow the spread of the infection until the end of April 2020 [24]. Later on a number of new cases were increasingly reported at the national level. In response, the Ethiopian government has organized and deployed COVID-19 prevention and control task forces that structured from a central to a local level [25]. Similarly, as a member of that task force, Wolaita Sodo University has participated in the study area's COVID-19 awareness campaign [26].

However, the experts' observational findings discovered that during the initial stage of the pandemic, the community was strictly exercising the public health measures which were gradually disappeared afterward. Besides, the Ethiopian community tends to provide more credit to the spiritual explanation of health issues than the biomedical model [27]. Moreover, partly

due to cultural values the society did not comply with health professionals and official prescriptions and advice related to COVID-19 preventive measures [28]. Furthermore, nationally 23.96 million internet users, 44.86 million mobile connections, and 6.70 million social media users (of which 96.2% accessing via mobile) were reported in January 2021 [29]. Here, the relative contribution of dissemination of false news and information should not be underestimated. These may indicate that factors that may contribute to the adoption of COVID-19 preventive measures are still complex.

In Ethiopia, several studies were conducted to assess the knowledge, attitude, and practice toward COVID-19 preventive measures among individuals with chronic medical conditions. In the meantime, being male, being unmarried, no formal education, rural residence, income of less than 221 US Dollar, household family size greater than or equal to 4, poor knowledge of COVID-19, and poor attitude towards COVID-19 were the factors found significantly associated with lower COVID-19 prevention practice [12, 13, 30–32].

However, since the recognition of perceived health beliefs and practices is important for developing effective COVID-19 health intervention strategies, the health belief model (HBM) should have been investigated to understand patients' compliance levels with COVID-19 preventive practices [33]. Besides, to assess the patient's belief about; the chances of experiencing a risk or getting a condition or disease, how serious a condition and its consequences, the efficacy of the advised action to reduce risk or the seriousness of impact, the tangible and psychological costs of the advised action, their readiness and confidence to take the advised action the HBM constructs should have been used [34]. Therefore, perceived susceptibility, perceived severity, perceived benefits, perceived barriers, cues to action, and self-efficacy of the patients was examined to identify predictors of compliance level with COVID-19 preventive measures.

In addition, access to water and sanitation status of the household that could influence basic prevention practices of patients (e.g. may help to wash hands with soap frequently) should have been investigated to identify predictors of compliance level with COVID-19 preventive measures and to highlight access to water and sanitation related gaps in the study area. Similarly, home environment status indicators such as access to refrigeration, electricity, and any internet that could influence the feasibility of social distancing (e.g. help to stay at home by avoiding frequent visits to shops) should have been studied to identify predictors of the patient's compliance level with COVID-19 preventive measures [24].

When the ordinal outcome variable is generated from ordinal data with a stepping pattern, using ordinal (proportional odds) model with a specific link function is an informative and powerful method of analysis than multinomial model. Similarly, instead of a binary logistic model by using proportional odds model the loss of information that could occur due to the dichotomization of the outcome variable was minimized. Moreover, proportional odds help to find out a cumulative probability for each level of the ordinal responses [35].

Therefore, this study aimed to determine the level of compliance with COVID-19 preventive measures and identify its predictors among patients with common chronic diseases in public hospitals of Southern Ethiopia by applying the proportional odds model.

## Methods and materials

### Study design, setting and period

Between February and March 2021, a facility-based cross-sectional study design was employed in Wolaita Sodo University Comprehensive Specialized Hospital (WSUCSH) and Dawuro Tarcha General Hospital (DTGH), Southern Ethiopia. WSUTRH is found in Wolaita Sodo town, the administrative center of the Wolaita Zone of the Southern Nation, Nationalities, and People's Region (SNNPR). DTGH is found in Tarcha town, the administrative centre for the

Dawuro zone of the SNNPR. Both are the only public hospitals in the respective zones that have specialty follow-up clinics where patients with hypertension, diabetes mellitus, and bronchial asthma have been getting service for more than five years. The hospitals offer follow-up service three days a week, while an estimated daily patient flow for hypertension, diabetes and bronchiole asthma follow up was 79 and 41 for WSUCSH and DTGH, respectively.

## Participants

All patients with chronic diseases (Hypertension, Diabetic Mellitus, and Bronchial Asthma) who were on follow-up in the hospitals of Wolaita, and Dawuro zones were the source population. All patients with the chronic diseases who were on follow-up in WSUCSH and DTGH during the data collection period, and who fulfilled the inclusion criteria were the study population. Patients who were 18 years or older, who had hypertension, diabetic mellitus, and bronchial asthma and they were on follow-up in the hospitals during the data collection period were included in the study. Whereas patients unable to communicate via any channel, and admitted were excluded.

## Sample size determination

The sample size was calculated using a single population proportion formula with the following assumptions; 95% confidence level, 0.05 margin of error, and 50% proportion of compliance with COVID-19 preventive measures. After adding a 10% non-response rate, the determined final sample size was 423.

## Sampling technique and procedure

Initially, the calculated sample size was proportionally allocated across the two hospitals based on estimated daily average hypertensive, diabetic and asthmatic outpatient flow. Then, each hospital's allotted sample size was again proportionally divided across the corresponding outpatient clinics for hypertension, diabetes, and bronchial asthma based on the estimated patient number. A systematic random sampling method was used to interview the eligible patients. First, the sampling interval (K) was separately calculated for each outpatient clinic by dividing the total number of patients registered for follow-up by the allocated number of patients. Then, the lottery method was used to select the first sample from the sampling interval. Finally, next to the first sample the eligible patients were interviewed at regular intervals on the date of follow-up.

## Data collection methods and quality assurance

The data were collected by using ODK Collect which is an open-source Android mobile application. Data collectors and supervisors with health backgrounds were recruited, and training was provided on how to get the blank forms, fill the blank forms, and send the finalized forms by using the Android mobile application. In addition, interview techniques and ethical issues were also addressed during the training session. Before the actual data collection, a pre-test was done outside the study area with a population of similar characteristics using 5% of the total sample size. The study participants were interviewed after the follow-up care in the quiet room with COVID-19 precautions. During an exit interview, responses to the questions were validated, restricted, and labeled require because expressions such as constraint, relevant, and requirements were added to the data (S1 File).

## Instrument and measurements

A pre-tested, structured, and interviewer-administered questionnaire was used to collect the data. The questionnaire had six sections: the patient's compliance level with COVID-19 preventive measures which was used as the ordinal outcome variable; socio-demographic variables, including clinical characteristics, knowledge of the mode of transmission of COVID-19, attitude towards COVID-19 control, HBM constructs, and access to water and sanitation status of households including the patient's home environment status indicators were used as the explanatory variables.

1. Compliance level questionnaires: taken from prior study [36]. It contains 11 items to measure compliance level of patients with COVID-19 preventive measures. These items were prepared in the form of a 4-point scale response (1 = not at all, 2 = very little, 3 = somewhat, 4 = to a great extent). Overall score was calculated by adding each score up to 44. Then, the total scores ranged from 11 to 44 were obtained. Finally, using Bloom's cutoff point the score was categorized as low, medium, and high for less than 60% (11–21 score), 60–79% (22–27 score), and greater than or equal to 80% (28–44 score), respectively.

2. Access to water, sanitation, refrigeration, electricity, and internet of the household related questions were prepared based on a prior evidence [24].

3. The knowledge level of the patient towards the mode of transmission of COVID-19 was measured by using a questionnaire adopted from the Cameroon study [37]. It contains 7 questions that were answered on a True/False basis with an additional "I don't know" option. Then, the correct answers were assigned 1 point, while an incorrect/unknown answer was assigned 0 points. Finally, the total knowledge score points ranged from 1 to 7 were obtained and then categorized based on Bloom's cutoff point. Accordingly, the higher total score greater than or equal to 80% (6–7 score) and the lower total score less than 80% (1–5 score) were categorized as good and poor knowledge, respectively.

4. The patient's attitude towards COVID-19 control was measured by using 2 questions adopted from the China study [38]. The participant's agree/yes answer was assigned 1 point, while disagree/no/ I don't know answer was assigned 0 points. Total scores ranged from 0 to 2 were obtained, the highest score (2 point) indicating positive attitude towards COVID-19 control.

5. HBM constructs questionnaire: It contains 21 items that measured six constructs. Specifically, three different items were used to measure perceived susceptibility, perceived severity, perceived benefits, and perceived barriers separately, while four and five items were used to measure cues to action, and self-efficacy of the patients. Accordingly, a five point Likert scale response ranging from "Strongly disagree"(1) to "Strongly Agree" (5) were prepared for individual items. Finally, total score was separately calculated for each constructs by adding the respective item's score. This questionnaire was attested for content validity by health education and public health experts who are currently employed as faculty at Wolaita Sodo University, College of Health Sciences and Medicine. Besides, the internal consistency of the items used to measure the HBM constructs were evaluated by a Cronbach's alpha test. Accordingly, the test values were 0.91, 0.96, 0.95, 0.93, 0.77, and 0.84 for perceived susceptibility, perceived severity, perceived benefits, perceived barriers, cues to action, and self-efficacy, respectively.

## Data management and analysis procedure

After cleaning the data, descriptive statistics such as absolute and relative frequency were determined for categorical variables, whereas mean (SD) and median (IQR) were determined

for continuous variables, to describe the study participants. The analysis was performed by SPSS version 25.

The ordinal outcome variable (compliance level) was generated from ordinal data which were initially discrete in nature with a stepping pattern and subsequently grouped into ordered categories: low = 1, medium = 2, and high = 3. However, for the sake of making the interpretation logical, the earlier ordered categories were reversed to high = 1, medium = 2, and low = 3.

Since our outcome variable is measured at ordinal level, we have chosen an ordinal regression model to identify the independent predictors of compliance level. However, evidences have suggested and used additional assumptions that need to be fulfilled before running the model, so as to have a valid result [39–43]. Accordingly, independent variables should only be treated as either categorical or continuous variable which was done in our study. Similarly, multicollinearity among the explanatory variables was assessed by a variance inflation factor (VIF) value less than 10 cut off point, which was not a problem (see Table 4). Besides, the assumption of proportional odds was assessed to choose between proportional odds model and partial proportional odds model using a Full Likelihood Ratio test. The test of parallel lines (score test) output result declared that the proportional odds model was plausible in our study with ($\chi^2$(26) equal to 25.679, p-value equal to 0.481). In addition, a link function that appropriately fit the model was assessed using bar charts. The results of the chart showed a negatively skewed distribution of compliance level (Fig 1). Therefore, a complementary log-log link function is best to fit the model.

Then, proportional odds model using complementary log-log link function was carried out to identify factors that were predicted the compliance level. First, a bivariate proportional odds model was performed to assess the crude association between compliance level with COVID-19 preventive measures and individual explanatory variables at a p-value less than 0.25 (see Table 4). Next, multivariable proportional odds model was carried out to determine the independent predictors of compliance level (see Table 5).

In the meantime, the adequacy of the final model was assessed by using the Model Fitting Information, the Goodness-of-Fit, and the Nagelkerke Pseudo $R^2$. Accordingly, the Model Fitting Information's output result showed a significant improvement in the fit of the final model over to the baseline intercept-only model with ($\chi^2$(26) equal to 118.182, p-value less than 0.001). The result of Goodness of Fit indicated that the observed data fitted very well with our built model ($\chi^2$(734) equal to 714.264, p-value equal to 0.692), here we reported the deviance chi-square result because in our study most cells were sparse with zero frequencies in the 762 (66.7%). The Nagelkerke Pseudo $R^2$ test result indicated that 28.6% of the variance of compliance level is accounted for by the final model.

To facilitate interpretation, the regression coefficients of the final model were exponentiated to determine odds ratio and its 95% confidence interval (CI). Finally, results of the study are presented in the forms of adjusted odds ratio along with its 95%CI to declare the strength of association. Moreover, statistical significance for the final model was set at p-value less than 0.05.

## Ethical approval and consent to participate

Ethical approval letter was obtained from the Wolaita Sodo University College of health sciences and medicine's ethical review committee with the CHSM/ERC/9 reference number. The permits and support letter were given to WSUCSH and DTGH. The informed written consent was secured with study subjects before the commencement of the data collection. During the course of data collection, no financial provision was made and the rights or welfare of the study subjects was respected.

## Results

Out of the 423 study participants who were eligible for the study, 419 (99%) agreed to participate and gave response. Accordingly, data related to socio-demographic and clinical characteristics, knowledge of the mode of transmission of COVID-19 and access to water and sanitation status were collected from 419 study participants. In addition to the previously obtained data, participants who had information of the recommended COVID-19 preventive measures were asked to provide data related to the advised action. Consequently, 2 participants had no information about preventive measures that were recommended to adhere to), whereas 417 had. As a result, data related to their perceived health beliefs about the advised action, their attitude towards the advised action, and their level of compliance with the recommended preventive measures were collected from 417 study participants (Fig 1).

### Socio-demographic and clinical characteristics

The median (IQR) age of the study participants was 45 (36–58) years. Majority of them were urban dwellers 315 (75.2%), married 297 (70.9%), completed tertiary education 197 (47%),

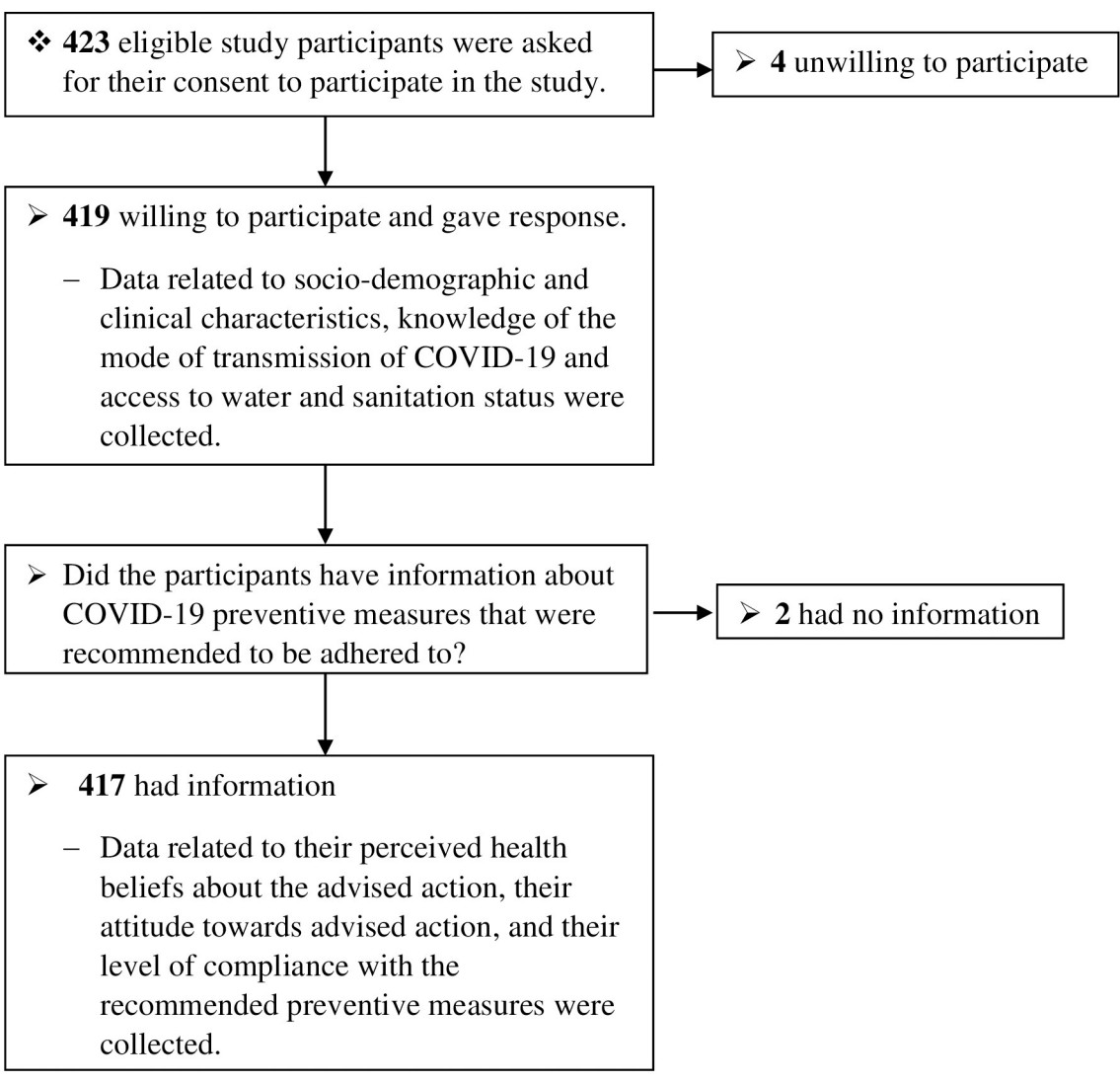

**Fig 1. Flow diagram showing the recruitment process of study participants.**

and were employed 282 (67.3%). In this study, television was mentioned as a major 362 (86.8%) source of information about COVID-19 preventive measures, while the website was mentioned as the least 96 (23.0%). Of all participants, half 214 (51.1%) had more than one type of chronic disease, and more than one-fifth (22.2%) utilized follow-up care for greater than or equal to 5 years (Table 1).

## Knowledge towards mode of transmission of COVID-19

Our study found that more than two-thirds (72.6%, 95%CI: 68.0%-76.8%) of the participants had poor knowledge of the mode of transmission of COVID-19. Respectively, seventy-seven

**Table 1. Socio-demographic and clinical characteristics of the study participants (n = 419).**

| Variables | Category | Frequency | Percent |
|---|---|---|---|
| Age (years) | 18–39 | 150 | 35.8 |
|  | 40–59 | 184 | 43.9 |
|  | > = 60 | 85 | 20.3 |
| Residence | Urban | 315 | 75.2 |
|  | Rural | 104 | 24.8 |
| Sex | Male | 224 | 53.5 |
|  | Female | 195 | 46.5 |
| Marital status | Married | 297 | 70.9 |
|  | Single | 64 | 15.3 |
|  | Divorced | 33 | 7.9 |
|  | Widowed | 25 | 6.0 |
| Completed educational level | Unable to read and write | 42 | 10.0 |
|  | Able to read and write | 16 | 3.8 |
|  | First cycle[A] | 17 | 4.1 |
|  | Second cycle[B] | 35 | 8.4 |
|  | High school and Preparatory[C] | 112 | 26.7 |
|  | Tertiary education[D] | 197 | 47.0 |
| Main work status over the past 12 months | Employed[E] | 282 | 67.3 |
|  | Unemployed[F] | 137 | 32.7 |
| Average monthly income in $[G] | < 254 | 372 | 88.8 |
|  | > = 254 | 47 | 11.2 |
| Sources of information about preventive measures (1,195)[H] | Television | 362 | 86.8 |
|  | Radio | 308 | 73.9 |
|  | Health workers advice | 274 | 65.7 |
|  | Written materials | 155 | 37.2 |
|  | Website | 96 | 23.0 |
| Number of chronic disease | One type | 205 | 48.9 |
|  | More than one type | 214 | 51.1 |
| Length of follow up in years | < 5 year | 326 | 77.8 |
|  | > = 5 year | 93 | 22.2 |

[A]grade1-3

[B]grade 4–8

[C]grade 9–12

[D]above grade 12.

[E]government, self, non-government.

[F]homemaker, retired, student, non-paid, able to work, unable to work.

[G]US Dollar ($) is converted from ETB based on average exchange rate of February and March 2021.

[H]due to multiple responses the sum becomes greater than the sample size.

**Table 2. The participants' knowledge towards COVID-19 mode of transmission.**

| How is COVID 19 transmitted? | True (%) | False (%) | I don't know (%) |
|---|---|---|---|
| Droplets when an infected person coughs, sneezes or speaks | 410 (97.9) | 6 (1.4) | 3(0.7) |
| Kissing an infected person | 382 (91.2) | 32 (7.6) | 5 (1.2) |
| Handshake | 394 (94.0) | 21 (5.0) | 4 (1.0) |
| Touching a contaminated surface and then touching your eyes, nose or mouth | 311 (74.2) | 81 (19.3) | 27 (6.4) |
| Blood transfusion[F] | 77 (18.4) | 201(48.0) | 141(33.7) |
| Sexual intercourse[F] | 182 (43.4) | 128 (30.5) | 109 (26.0) |
| Contaminated foodstuffs | 115 (27.4) | 164 (39.1) | 140 (33.4) |
| **Overall knowledge level** | **Good (%)** = 115 (27.4), **Poor (%)** = 304 (72.6) | | |

F: false answer.

(18.4%), and 182 (43.4%) participants responded to blood transfusion and sexual intercourse as a means of COVID-19 transmission which was a false answer (Table 2).

## Health belief model constructs for COVID-19

Of the total studied participants, 417 who had information about COVID-19 preventive measures were asked to indicate their level of agreement on a list of items. More than half of the participants (59.0%) agreed with worrying a lot about getting the disease, while 51.6% of participants disagreed with searching for new information to know how to prevent the disease. The overall Median (IQR) score of perceived susceptibility to the infection, and cues to action among the participants were 11.00 (8.00–12.00), and 12.00 (8.50–14.00), respectively (Table 3).

**Table 3. The study participants' perceived susceptibility and cues to action.**

| Items | [A]Disagree (%) | Neutral (%) | [B]Agree (%) |
|---|---|---|---|
| [1]My medical conditions make me more likely that I will get the disease | 134 (32.1) | 116 (27.8) | 167 (40.0) |
| [1]I feel that my chances of getting the disease in the future is high | 89 (21.3) | 136 (32.6) | 192 (46.0) |
| [1]I worry a lot about getting the disease | 90 (21.6) | 81 (19.4) | 246 (59.0) |
| **Overall perceived susceptibility, Median (IQR) = 11.00 (8.00–12.00)** | | | |
| [2]I search for new information to know how to prevent the disease | 215 (51.6) | 56 (13.4) | 146 (35.0) |
| [2]I always follow medical orders to prevent myself from the disease | 177 (42.4) | 116 (27.8) | 124 (29.7) |
| [2]I take vitamins and vegetables to prevent the virus | 123 (29.5) | 109 (26.1) | 185 (44.4) |
| [2]I do exercise at least three times a week | 176 (42.2) | 71 (17.0) | 170 (40.8) |
| **Overall cues to action, Median (IQR) = 12.00 (8.50–14.00)** | | | |

[1] susceptibility item

[2] cues to action item.

[A] the merge of strongly disagree and disagree

[B] the merge of agree and strongly agree.

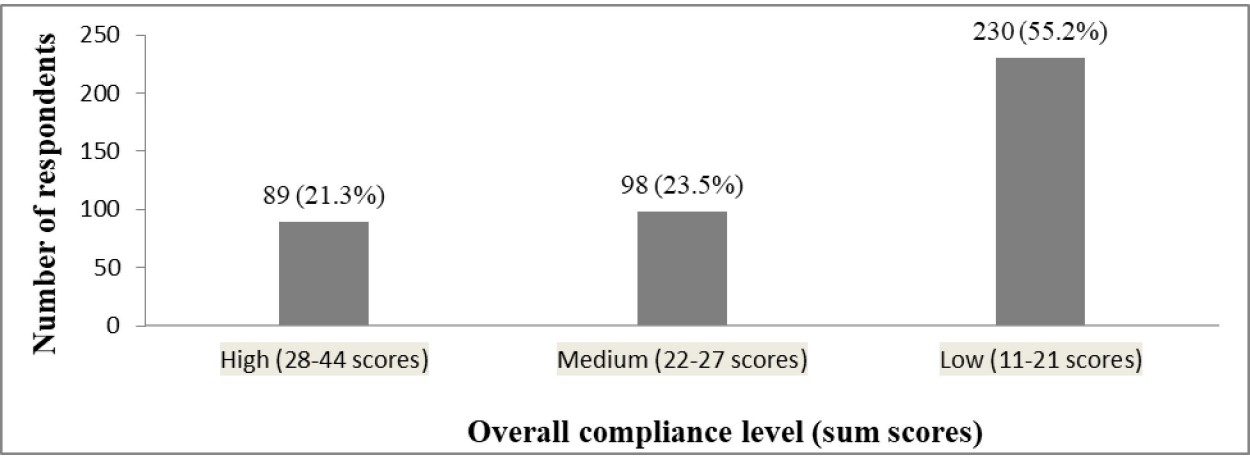

**Fig 2. The participants' overall compliance level with COVID-19 preventive measures.**

### Compliance level of the study participants

Our findings revealed that, respectively, 89 (21.3%), 98 (23.5%), and 230 (55.2%) participants had high, medium, and low compliance levels with COVID-19 preventive measures. Similarly, we found overall compliance sum scores that ranged from 11 to 44. Of all participants, only 11 (2.6%) scored the maximum score (i.e. 44) (Fig 2).

### Bivariate proportional odds model and multicollinearity diagnosis

Table 4 shows, candidate explanatory variables that had a p-value less than 0.25 in the bivariate proportional odds model analysis and had VIF less than 10 in the multicollinearity diagnosis. As a result, the variables were taken into a final model to determine the independent predictors of compliance level.

### Predictors of compliance level with COVID-19 preventive measures

According to the multivariable proportional odds model result, perceived susceptibility to the infection (AOR: 0.91, 95%CI: 0.84, 0.97), cues to action, or being ready to practice preventive measures (AOR: 0.89, 95%CI: 0.85, 0.94), having access to drinking water that piped into the dwelling (AOR: 0.52, 95%CI: 0.32, 0.84), and having no access to any internet in the last 12 months (AOR: 0.62, 95%CI: 0.42, 0.92) showed statistically a significant lower cumulative odds of having a low compliance level with COVID-19 preventive measures, whereas not having a functional refrigerator in the house (AOR: 2.17, 95%CI: 1.26, 3.74), and having poor knowledge towards COVID-19 mode of transmission (AOR: 1.42, 95%CI: 1.02, 1.98) showed statistically a significant higher cumulative odds of having low compliance level, keeping all other variables constant (Table 5).

### Discussions

Our study determined more than half of the study participants had low compliance levels with COVID-19 preventive measures in the study area. In the meantime, a multivariable proportional odds model identified that perceived susceptibility to the infection, cues to action or being ready to practice preventive measures, having access to drinking water piped into the dwelling, and having no access to any internet in the last 12 months showed a significantly lower cumulative odds of having low compliance level with COVID-19 preventive measures,

**Table 4. Bivariate proportional odds model and multicollinearity diagnosis results (n = 417).**

| Variables | Categories | P-value | VIF |
|---|---|---|---|
| Residence | Urban (315 (75.5%))* | < 0.001 | 1.832 |
| | Rural (102 (24.5%))* | | |
| Sex | Male (223 (53.5%))* | 0.061 | 1.184 |
| | Female (194 (46.5%))* | | |
| Marital status | Married (296 (71.0%))* | 0.246 | 1.168 |
| | Others[A] (121 (29.0%))* | | |
| Completed educational level | No formal education[B] (56 (13.4%))* | 0.004 | 1.631 |
| | Formal education[C] (361 (86.6%))* | | |
| Main work status over the past 12 months | Employed (282 (67.6%))* | 0.140 | 1.232 |
| | Unemployed (135 (32.4%))* | | |
| Average monthly income in $[D] | < 254 (370 (88.7%))* | < 0.001 | 1.430 |
| | > = 254 (47 (11.3%))* | | |
| Mention television as source of information | No (55 (13.2%))* | 0.001 | 1.681 |
| | Yes (362 (86.8%))* | | |
| Mention written materials as source of information | No (262 (62.8%))* | 0.005 | 1.960 |
| | Yes (155 (37.2%))* | | |
| Mention website as source of information | No (321 (77.0%))v | 0.010 | 1.565 |
| | Yes (96 (23.0%))* | | |
| Overall knowledge level towards COVID-19 mode of transmission | Poor (302 (72.4%))* | 0.002 | 1.225 |
| | Good (115 (27.6%))* | | |
| Overall attitude towards COVID-19 control | Negative (313 (75.1%))* | 0.002 | 1.224 |
| | Positive (104 (24.9%))* | | |
| Overall perceived susceptibility | 11.00 (8.00–12.00)[¥] | < 0.001 | 2.092 |
| Overall perceived severity | 11.39 (±3.76)[£] | 0.004 | 2.278 |
| Overall perceived benefits | 12.00 (9.00–12.00)[¥] | < 0.001 | 1.831 |
| Overall perceived barriers | 12.00 (12.00–14.00)[¥] | 0.007 | 1.217 |
| Overall cues to action | 12.00 (9.00–14.00)[¥] | < 0.001 | 1.580 |
| Overall self-efficacy | 18.00 (15.00–20.00)[¥] | < 0.001 | 1.925 |
| Source of drinking water for the HHs | Water piped into dwelling (182 (43.6%))* | < 0.001 | 2.345 |
| | Others [E] (235 (56.4%))* | | |
| Distance of water source in min | 10.00 (4.00–20.00)[¥] | 0.025 | 2.287 |
| Access to hand washing facility | No (272 (65.2%))* | 0.001 | 2.796 |
| | Yes (145 (34.8%))* | | |
| Use soap for washing hands | No (321 (77.0%))* | 0.002 | 2.446 |
| | Yes (96 (23.0%))* | | |
| Size of the family members in the HHs | 4.98 (±2.07)[£] | 0.164 | 1.454 |
| Number of rooms in house | 3.79 (±1.69)[£] | 0.147 | 1.424 |
| Have functional refrigerator in the house | No (176 (42.2%))* | < 0.001 | 2.795 |
| | Yes (241 (57.8%))* | | |
| Have access electricity in house | No (109 (26.1%))* | 0.025 | 2.125 |
| | Yes (308 (73.9%))* | | |
| Have access to internet in the last 12 months | No (252 (60.4%))* | 0.004 | 1.889 |
| | Yes (165 (39.6%))* | | |

VIF: Variance Inflation Factor.

[A] single, divorced, widowed.

[B] unable read and write, able read and write but no formal education.

[C] first cycle, second cycle, high school and preparatory, tertiary education.

[D] US Dollar ($) is converted from ETB based on average exchange rate of February and March 2021.

[E] water piped into yard/plot, using a public tap or standpipe.

*number (marginal percentage)

[¥] median (IQR)

[£] Mean (±SD)

**Table 5. Multivariable proportional odds model for predictors of compliance level.**

| Variables | | Estimate (SE) | P-value | AOR (95%CI) |
|---|---|---|---|---|
| Threshold | [High = 1] | -3.292 (0.993) | 0.001 | |
| | [Moderate = 2] | -2.225 (0.987) | 0.024 | |
| Residence (Urban vs. Rural) | | 0.003 (0.256) | 0.990 | |
| Sex (Male vs. Female) | | -0.181 (0.164) | 0.271 | |
| Marital status (Married vs. Others) | | -0.219 (0.187) | 0.241 | |
| Completed educational level (No Formal education vs. Formal education) | | -0.113 (0.327) | 0.731 | |
| Main work status over the past 12 months (Employed vs. Unemployed) | | 0.104 (0.182) | 0.569 | |
| Average monthly income (< $254 vs. > = $254)[A] | | 0.405 (0.257) | 0.114 | |
| Mention television as source of information (No vs. Yes) | | 0.591 (0.367) | 0.107 | |
| Mention written materials as source of information (No vs. Yes) | | -0.044 (0.211) | 0.837 | |
| Mention web-sites as source of information (No vs. Yes) | | -0.189 (0.215) | 0.379 | |
| Overall knowledge level towards COVID-19 mode of transmission (Poor vs. Good) | | 0.350 (0.171) | **0.041** | **1.42 (1.02, 1.98)** |
| Overall attitude towards COVID-19 control (Negative vs. Positive) | | 0.276 (0.184) | 0.132 | |
| Overall perceived susceptibility | | -0.099 (0.036) | **0.006** | **0.91 (0.84, 0.97)** |
| Overall perceived severity | | 0.053 (0.031) | 0.088 | |
| Overall perceived benefits | | 0.001 (0.036) | 0.983 | |
| Overall perceived barriers | | 0.036 (0.026) | 0.161 | |
| Overall cues to action | | -0.115 (0.026) | < 0.001 | **0.89 (0.85, 0.94)** |
| Overall self-efficacy | | -0.046 (0.029) | 0.117 | |
| Source of drinking water for the HHs (Water piped into dwelling vs. Others [B]) | | -0.651 (0.242) | **0.007** | **0.52 (0.32, 0.84)** |
| Distance of water source from the house in min | | -0.006 (0.009) | 0.542 | |
| Have access to hand washing facility (No vs. Yes) | | -0.052 (0.255) | 0.839 | |
| Use soap for washing hand (No vs.Yes) | | 0.301 (0.261) | 0.249 | |
| Size of the family members in the HHs | | 0.031 (0.046) | 0.503 | |
| Number of rooms in house | | 0.005 (0.052) | 0.929 | |
| Have functional refrigerator in the house (No vs. Yes) | | 0.776 (0.277) | **0.005** | **2.17 (1.26, 3.74)** |
| Have access electricity in house (No vs. Yes) | | -0.471 (0.283) | 0.096 | |
| Have access to internet in the last 12 months (No vs. Yes) | | -0.475 (0.201) | **0.018** | **0.62 (0.42, 0.92)** |

[A]US Dollar ($) is converted from ETB based on average exchange rate of February and March 2021.

[B]water piped into yard/plot, using a public tap or standpipe

NB. The last group is used as a reference.

whereas having no functional refrigerator in the house and having poor knowledge towards COVID-19 mode of transmission showed a significantly higher cumulative odds of having low compliance level.

In our study, almost all (417 out of 419, 99.5%) participants had information about COVID-19 preventive measures; of these, the majority (362 out of 417, 86.8%) of them mentioned television as their primary source of information. Similarly, the majority of the participants were urban dwellers (75.2%) and married (70.9%). Analogous to previous findings [12, 13, 30, 31].

Besides, we found that more than two-thirds (72.6%, 95%CI: 68.0%-76.8%) of the participants had poor knowledge about the mode of transmission of COVID-19, it was categorized

based on Bloom's cutoff point less than 80% scored points. This finding was considerably higher than nationally conducted study findings, while it was consistent with study findings from the Democratic Republic of Congo and Northern Nigeria which revealed (66.1%), (65.4%), (62.41%), (55.4%), (70.0%), (69.53%) [12, 13, 30, 31, 44, 45]. The detected national level discrepancy might be related to differences in content and dimension of the questions that were used to measure the participants' knowledge. However, a higher discrepancy was observed with study from Vietnam (31.6%) [46], which might be due to the difference in the study setting. Unexpectedly, almost one-fifth (18.4%) of the participants responded blood transfusion was a mean of COVID-19 transmission, whereas 43.4% response that sexual intercourse was another way of transmission in our study. Other studies from Ethiopia and elsewhere showed that 12.38% and 7.7% of participants responded blood transfusion was a mean of COVID-19 transmission, while 9% and 7.4% of participants answered sexual intercourse was another mean [37, 47]. This might need the attention of program providers, and health professionals to clear the misunderstanding of participants regarding the COVID-19 mode of transmission.

We conducted this study a year after the first case was identified at the national level and after the government had deployed COVID-19 prevention and control task forces from a central to a local level. However, our findings revealed that more than half (55.2%) of the participants had low compliance levels in the study area with 95%CI (50.4%-59.9%). This finding is nearly consistent with study findings from Ethiopia that reported 55.2% and 49.6% [48, 49]. This unexpected low compliance level in the study area could verify the experts' observation that revealed a growing community ignorance of the COVID-19 preventive measures. Besides, this finding might be related to the evidence indicating that the Ethiopian community tends to provide more credit to the spiritual explanation of health issues than the biomedical model [27]. Similarly, a qualitative study conducted in northwest Ethiopia explored strong cultural and religious practices as one of the major perceived barriers to COVID-19 prevention practices [50]. In addition, other qualitative study also revealed that some participants have linked the disease with divine power [26]. Moreover, partly due to cultural values, particularly social solidarity groups such as "Equb" (a traditional means of saving in Ethiopia), "Iddir" (a traditional group formed to support its members during bereavement), and funereal ceremonies; and religious practices the society did not comply with health professionals and official prescriptions and advice related to COVID-19 preventive measures. Therefore, our finding suggests the significance of integrating the Mass Media, particularly the communities' primary source of information (e.g., in this study television); religious leaders; cultural values such as "Equb", "Iddir", and funereal ceremonies to convey health information which in turn help to adopt inclusive and effective preventative health behavior at the local level [26–28, 48].

HBM related predictors, the median score result of our study indicated that half of all the observed overall perceived susceptibility scores and overall cues to action scores of the participants were less than 11.00 and 12.00, respectively. Also, the level of the scores showed a statistically significant effect on the participants' compliance level with COVID-19 preventive measures. Accordingly, with one unit increase in the perceived susceptibility score of the participants, the cumulative odds of having a low compliance level is 0.91 times lower as compared with a high compliance level. It means, the probability of having a higher compliance level was found to increase with the increase in the participant's perceived susceptibility to the infection. In support of our study, the previous community-based study findings of Ethiopia and elsewhere point out that perceived susceptibility has shown a significant positive association with the compliance level of participants [48, 51]. Similarly, other evidence also shows that for an individual to practice preventive behaviors, he/she needs to believe he/she is personally susceptible to such health problem [52].

Also, prior studies from Ethiopia and Saudi Arabia [33, 51, 53] reported a statistically significant positive association between the participant's cues to action and the adoption of COVID-19 preventive behaviors. In line with these reports, our study found that for every one-unit increase in the cues to action score of the participant, the log odds of having a lower compliance level would be decreased by 0.115 and the associated odds ratio is 0.89. This implies the more individuals are ready to practice preventive measures, the more likely to have a higher compliance level. This implies the noteworthy of improving the participants' readiness levels to better adapt to the recommended preventive measures. In contrast to other community-based studies [48, 53], perceived benefit, perceived barrier, and self-efficacy were not associated with the compliance level in our study. This might be related to the fact that the adoption of preventive health behavior depends on the types and accuracy of risk perceptions which might vary according to gender, age, education, place of residence, and the set of social beliefs [52].

Our study showed that the estimated cumulative odds of having a low compliance level with the COVID-19 preventive measures were about one and half times higher among participants who have poor knowledge of the COVID-19 mode of transmission as compared to their counterparts, which is supported by a study conducted in Ethiopia [47]. Others have shown that individuals who had poor COVID-19 related knowledge were less likely to practice personal preventive measures [12, 13, 31]. This could simply indicate that it is important to design and use different strategies to improve the participants' basic knowledge of COVID-19, to probably help the participants to improve their compliance level [27].

Regarding access to water, and home environment related predictors, despite the fact that the evidence revealed the provision of water is essential to ensure good and consistent application of sanitation, and hygienic practice in a home, which would help to prevent human-to-human transmission of the COVID-19 [24, 28, 54]. However, our study found that more than half (56.4%) of participants had no access to drinking water that was piped into a dwelling, resulting in a low compliance level in the study area. Accordingly, we found that participants having access to drinking water that was piped into the dwelling had a 52% lower cumulative probability of having a low compliance level with COVID-19 preventive measures compared to those who have no access to drinking water that was piped into a dwelling. Therefore, governmental and non-governmental organizations should address the identified gaps to improve participants' compliance levels in the study area.

Additionally, the ordered odds of participants who have no functional refrigerator in the house were two times more likely to be in low compliance level as compared to those who have a functional refrigerator in the house. This might be related to the evidence that showed access to refrigeration helps family members to avoid frequent visits to shops, then enable them to stay at home [24].

Moreover, patients who had no access to any internet in the last 12 months were 62% less likely to have a lower compliance level to COVID-19 preventive measures compared to those who had access. This finding could tie with a previous finding that revealed the participants who received COVID-19 information from social media were less likely to adhere to COVID-19 preventive measures [55]. Besides, our finding supports the notion that the dissemination of false news and information without scientific nature, could hamper the adoption of preventive behaviour [52]. Similarly, our result supported the experts' view that revealed misleading and contradicting information coming from the internet could lead the individual to ignorance of the recommended preventive measures [27]. Therefore, a piece of health information and the communication strategy should be designed, planned, and implemented in a way that could minimize misinformation surrounding the COVID-19 disease, and similar future outbreaks related to the emergence of new variants.

### Implications for clinical practice and future research

The determination of the level of compliance with the covid19 preventive measures and its predictors in patients with chronic illnesses are a valuable contribution that will help health care planners, program providers, health professionals, and policymakers to design, plan and implement new interventional strategies. Moreover, the information could also help to revise the previous governmental interventions in a way that improves participants' compliance level and control similar future outbreaks related to the emergence of new COVID-19 variants. Furthermore, it is essential to know the availability of enabling conditions before implementing any preventive measures in any setting.

### Strength and limitation of the study

To the best of our knowledge, this study is the first to use the proportional odds model; therefore the loss of information that could occur due to dichotomization of the outcome variable was minimized. This study has used a validated self-constructed scale to measure the outcome variable and the patient's knowledge level. Whereas the items of HBM constructs were tested for reliability, then the acceptable to the highest value of Cronbach alpha ranging from 0.772 to 0.958 was obtained. Moreover, the predictors' narrow confidence intervals observed in the final model reflect the high precision of the estimation along with a sufficient sample size.

However, our study has a number of limitations to consider. A social desirability bias might have been introduced because the respondents were asked to what extent they acted in accordance with the COVID-19 prevention guidelines. Besides, it is possible that the results cannot be applied to the two zones' wider populations because the study only included individuals who attended the follow-up clinics. In addition, because of the nature of the cross-section study, our study could not show the trends of compliance with COVID-19 preventive measures over time in the study setting. Furthermore, the generalizability of the findings may not be effective for the national wide because only chronic disease patients who had follow-up care in two zones of South Ethiopia were scrutinized.

## Conclusions

More than half of the study participants had low compliance levels with COVID-19 preventive measures. 'Perceived susceptibility to the infection, cues to action or being ready to practice preventive measures, having access to drinking water piped into the dwelling, having no access to any internet in the last 12 months, having no functional refrigerator in the house, and having poor knowledge of COVID-19 mode of transmission were found to be the independent predictors of low compliance level with COVID-19 preventive measures.

## Supporting information

**S1 Dataset. Predictors of compliance level dataset.**
(SAV)

**S1 File. English version an ODK excel form.**
(XLSX)

## Acknowledgments

We would like to extend our gratitude to the study participants, data collectors, supervisors, and administrators.

## Author Contributions

**Conceptualization:** Temesgen Bati Gelgelu.

**Data curation:** Temesgen Bati Gelgelu, Shemsu Nuriye, Tesfaye Yitna Chichiabellu, Amene Abebe Kerbo.

**Formal analysis:** Temesgen Bati Gelgelu.

**Funding acquisition:** Temesgen Bati Gelgelu.

**Investigation:** Temesgen Bati Gelgelu, Amene Abebe Kerbo.

**Methodology:** Temesgen Bati Gelgelu, Tesfaye Yitna Chichiabellu, Amene Abebe Kerbo.

**Project administration:** Temesgen Bati Gelgelu, Amene Abebe Kerbo.

**Resources:** Temesgen Bati Gelgelu, Tesfaye Yitna Chichiabellu, Amene Abebe Kerbo.

**Software:** Temesgen Bati Gelgelu.

**Supervision:** Temesgen Bati Gelgelu, Shemsu Nuriye, Tesfaye Yitna Chichiabellu, Amene Abebe Kerbo.

**Validation:** Temesgen Bati Gelgelu.

**Visualization:** Temesgen Bati Gelgelu, Shemsu Nuriye, Tesfaye Yitna Chichiabellu, Amene Abebe Kerbo.

**Writing – original draft:** Temesgen Bati Gelgelu.

**Writing – review & editing:** Temesgen Bati Gelgelu, Shemsu Nuriye, Tesfaye Yitna Chichiabellu, Amene Abebe Kerbo.

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
