## [Decision Letter · Decision Letter 0]

16 May 2022

PONE-D-22-10853Predictors of compliance level with COVID-19 preventive measures among patients with common chronic diseases in public hospitals of Southern Ethiopia: A proportional odds modelPLOS ONE

Dear Dr. Gelgelu,

Thank you for submitting your manuscript to PLOS ONE. After careful consideration, we feel that it has merit but does not fully meet PLOS ONE’s publication criteria as it currently stands. Therefore, we invite you to submit a revised version of the manuscript that addresses the points raised during the review process. Dear Author

Please find attached the results of your peer review process done by two independent reviewers. 

We are eager to received your feedback and comments and we appreciate a quick response. 

Best regards

**Juan A López-Rodríguez**

We look forward to receiving your revised manuscript.

Kind regards,

**Juan A López-Rodríguez**

*Academic Editor*

**PLOS ONE**

Journal Requirements:

"This study was funded by the Wolaita Sodo University. The funder had no role in the study design, data analysis, result interpretation, and decision to publish."

"This study was funded by Wolaita Sodo University with the wsu41/19/368 grant numbers. Received byTBG. The funders had no role in study design, data collection and analysis, decision to publish, or preparation of the manuscript."

Reviewers' comments:

Reviewer's Responses to Questions

**Comments to the Author**

1. Is the manuscript technically sound, and do the data support the conclusions?

Reviewer #1: Yes

Reviewer #2: Yes

2. Has the statistical analysis been performed appropriately and rigorously? 

Reviewer #1: I Don't Know

Reviewer #2: Yes

3. Have the authors made all data underlying the findings in their manuscript fully available?

Reviewer #1: Yes

Reviewer #2: Yes

4. Is the manuscript presented in an intelligible fashion and written in standard English?

Reviewer #1: Yes

Reviewer #2: Yes

5. Review Comments to the Author

Reviewer #1: Thought at the beginning I thought this paper wasn't interesting at all, I do believe it shows that knowing your population before taking any healthcare measure is a must. In Spain, when I ask my patients to wash frequently their hands I do not consider the possibility of not having access to a hand washing facility or if I give them written instructions to improve the preventive measure they can take to avoid covid I wouldn't think that 10% are unable to read. That's why I believe the authors conclusions are right but could be generalized to include something like "know your population before implementing any preventive measure"

Reviewer #2: I would like to acknowledge the effort of the authors to do this research. I agree this is an important topic to address especially as COVID-19 vaccination is not so high in Ethiopia and some African countries, we need to help patients to understand how to prevent COVID-19. Said that, I found confusing how they built the test and how they measured the different sections of the test. I think they have to make the information easier to read.

Title: I would recommend the short title adding the location.

Abstract: I would appreciate that they quote at least 1-2 preventive measures in the background to be able to fully understand their results.

Manuscript:

Lines 63-64, As the results are related with the socioeconomic condition of the patients, it would be interesting to add a sentence to describe as low socioeconomic status makes people more vulnerable to COVID-19.

Lines 88-90. It would be easier to understand the results if they described which category was more related to the preventive practices as they described the general characteristics (example: not sex but males, etc.).

Lines 119-124. This looks like a strategy for research or for practice in the future. I would add this sentence to the discussion.

I miss that the transmission of COVID-19 is described in the introduction, especially, because the airborne transmission has not been described in this article although many scientists agree about the airborne transmission and not only the droplets. https://pubmed.ncbi.nlm.nih.gov/33865497/

Some explanation about why the airborne transmission was not included in the text must be shared in the discussion.

Lines 128-134, please add the country. It would help to know the population that is assigned to the WSUCSH and DTGH, especially because after the sample size is calculated. If the facilities are public ones, it would be nice to say it here and not later in the text (137)

Also, as some patients have been chronic patients who were looked after more than 5 years, it would be nice to add it here.

Lines 136-141. I would simplify the text, a suggestion: “Inclusion criteria: patients who were > 18 years, they had chronic diseases (hypertension, diabetic mellitus, and bronchial asthma) and they were on follow-up in the hospitals during the data collection period. Whereas unable to communicate via any channel, and admitted patients were excluded”.

Lines 148-150 This information will be better located in the setting. On the other hand, are these clinics the only ones in the whole country, in the area? Please, clarify.

Lines 150-152, it would be clearer if they describe the flow of patients in the clinics in the setting section in line 128-134.

Lines 161-206. I found difficult to follow this section. I don’t know if I am wrong but I have understood that the authors created the test with sections of different surveys that have been administered in other studies. If this is the case, it would be nice to start with this clear information. I would suggest to add a table with the scores of each scale. It is difficult to interpret the results without a clear idea of how each variable was measured (169-206). Also finishing with the score of this final test.

Line 209. I would suggest to describe this section after sampling technique. I was confused reading line 162 about the pre-tested.

Line 292-296, As a suggestion: It would be interesting to know if patients who had several conditions are more prone to compliance the preventive measures.

The titles of the tables must be simplified. There is no need to describe the patients and the setting as both characteristics have been described in methods before.

In table 6, the average monthly income is difficult to understand for an international audience, please change it to USA dollars.

Lines 350-351 has already been said before.

Lines 357-362, it would be needed to compare with similar countries to make the text more interesting for an international audience.

Lines 370-375. I would like a little bit of context about the COVID-19 health promotion in Ethiopia. Was there any national campaign to explain the protection measures? What kind of education both hospitals offered to their patients? May be patients were not protecting themselves because anyone explained them clearly how to do it. As mass media or religious leaders seem to have a role in the community, it would be nice to quote if any of the leaders in the community or in the country took a proactive role to share preventive measures.

Lines 424-434. I think this is a key result, clean water is a basic need and this study gives another motive to support the access to clean water in each home.

Line 440-451. If we are going to discuss the impact of having internet access. There should be a reference in the introduction regarding the use of internet in Ethiopia. Secondly, it would be helpful to understand how people in the community are using it, do they use more social media (Facebook, WhatsApp, Telegram, etc.) than other sources of information?

6. PLOS authors have the option to publish the peer review history of their article (what does this mean?). If published, this will include your full peer review and any attached files.

Reviewer #1: **Yes: **Dr. Santiago Machin-Hamalainen, MD

Reviewer #2: No

---

## [Author Response · Author response to Decision Letter 0]

13 Jul 2022

We would like to thank the editors and the reviewers for careful review of our manuscript and providing us with their comments and suggestions to further improve the quality of the manuscript. Accordingly, we have prepared the following responses to address the editors` and the reviewers` comments and suggestions. 

Responses for editors` comments:

Journal Requirements: “when submitting your revision, we need you to address these additional requirements.”

Comment_1. Please ensure that your manuscript meets PLOS ONE's style requirements, including those for file naming. The PLOS ONE style templates can be found at 

Response_1. We have formatted the manuscript in accordance with PLOS ONE's style requirements, including file naming (Please, see the uploaded Revised Manuscript with Track Changes and the Manuscript file). 

Comment_2. Thank you for stating the following in the Acknowledgments Section of your manuscript: 

"This study was funded by the Wolaita Sodo University. The funder had no role in the study design, data analysis, result interpretation, and decision to publish." 

"This study was funded by Wolaita Sodo University with the wsu41/19/368 grant numbers. Received by TBG. The funders had no role in study design, data collection and analysis, decision to publish, or preparation of the manuscript." 

Response_2. We have removed any funding-related text from the manuscript (Please, see the acknowledgements and funding sections of Revised Manuscript with Track Changes file). So, please update our Funding Statement as follows: This study was funded by the Wolaita Sodo University with the wsu41/19/368 grant numbers. Received by TBG. The funders had no role in study design, data collection and analysis, decision to publish, or preparation of the manuscript.

Comment_3. We note that you have indicated that data from this study are available upon request. PLOS only allows data to be available upon request if there are legal or ethical restrictions on sharing data publicly. For more information on unacceptable data access restrictions, please see http://journals.plos.org/plosone/s/data-availability#loc-unacceptable-data-access-restrictions. 

Response_3. There are no restrictions on sharing our study`s data. Therefore, currently we have uploaded the minimal dataset necessary to replicate the study findings as Supporting Information files. Besides, we kindly request you to update the Data Availability statement on our behalf.

Responses for reviewers' comments:

“Reviewer's Responses to Questions”

1. Is the manuscript technically sound, and do the data support the conclusions?

Reviewer #1: Yes

Reviewer #2: Yes

Response. We appreciate that

2. Has the statistical analysis been performed appropriately and rigorously?

Reviewer #1: I Don't Know

Reviewer #2: Yes

Response. The statistical analyses were performed in accordance with PLOS ONE's SAMPL guidelines, the “Statistical Analyses and Methods in the Published Literature”. The required “General Principles for Reporting Statistical Methods” are followed. Accordingly; 

Preliminary analyses: Identifying any statistical procedures used to modify raw data before analysis such as collapsing continuous data into categorical data or combining categories or vice versa is done (Please, see the instrument and measurements, and data management and analysis procedure sections). 

Primary analyses: Describing the purpose of the analysis, summarizing data using descriptive statistics, naming the statistical package used, and reporting the level of significance is performed (Please, see the data management and analysis procedure section).

Supplementary analyses: Describing methods used for any ancillary analyses, such as testing of assumptions underlying methods of analysis is performed (Please, see the data management and analysis procedure section). 

3. Have the authors made all data underlying the findings in their manuscript fully available?

Reviewer #1: Yes

Reviewer #2: Yes

Response. We appreciate that

4. Is the manuscript presented in an intelligible fashion and written in standard English?

Reviewer #1: Yes

Reviewer #2: Yes

Response. We appreciate that

“5. Review Comments to the Author”

Comment. Please use the space provided to explain your answers to the questions above. You may also include additional comments for the author, including concerns about dual publication, research ethics, or publication ethics. (Please upload your review as an attachment if it exceeds 20,000 characters)

Reviewer #1: Thought at the beginning I thought this paper wasn't interesting at all, I do believe it shows that knowing your population before taking any healthcare measure is a must. In Spain, when I ask my patients to wash frequently their hands I do not consider the possibility of not having access to a hand washing facility or if I give them written instructions to improve the preventive measure they can take to avoid covid I wouldn't think that 10% are unable to read. That's why I believe the authors conclusions are right but could be generalized to include something like "know your population before implementing any preventive measure"

Response. We thank you so much for the acknowledgement and the suggestion. Accordingly, the suggested statement is incorporated in the conclusions section of the revised manuscript. 

Reviewer #2: I would like to acknowledge the effort of the authors to do this research. I agree this is an important topic to address especially as COVID-19 vaccination is not so high in Ethiopia and some African countries, we need to help patients to understand how to prevent COVID-19. Said that, I found confusing how they built the test and how they measured the different sections of the test. I think they have to make the information easier to read.

Response. We appreciate your acknowledgement and comment. In the current revised manuscript, we have revised the entire data management and analysis procedure section. Besides, we have moved all issues related with the tests from results section to data management and analysis procedure section. 

Comment. “Title: I would recommend the short title adding the location.”

Response. We accepted the comment and now, the title is modified accordingly to PLOS ONE's a full title requirements. 

Comment. “Abstract: I would appreciate that they quote at least 1-2 preventive measures in the background to be able to fully understand their results.”

Response. Now, some preventive measures are included accordingly in the abstract section; introduction subsection. 

Manuscript:

Comment. “Lines 63-64, As the results are related with the socioeconomic condition of the patients, it would be interesting to add a sentence to describe as low socioeconomic status makes people more vulnerable to COVID-19.”

Response. A sentence that describes the association between low socio-economic status and COVID-19 is included in the introduction section. 

Comment. “Lines 88-90. It would be easier to understand the results if they described which category was more related to the preventive practices as they described the general characteristics (example: not sex but males, etc.).”

Response. Now, the specific characteristics of the participants that associated with the preventive practices are described (Please, see the introduction section).

Comment. “Lines 119-124. This looks like a strategy for research or for practice in the future. I would add this sentence to the discussion.”

Response. We accepted the comment and the suggested modification are made (Please, see the first paragraph of discussion section). 

Comment. “I miss that the transmission of COVID-19 is described in the introduction, especially, because the airborne transmission has not been described in this article although many scientists agree about the airborne transmission and not only the droplets. https://pubmed.ncbi.nlm.nih.gov/33865497/

Some explanation about why the airborne transmission was not included in the text must be shared in the discussion.”

Response. Now, the evidences that related with mode of transmission of COVID-19 are included in the second paragraph of the introduction section.

Comment. “Lines 128-134, please add the country. It would help to know the population that is assigned to the WSUCSH and DTGH, especially because after the sample size is calculated. If the facilities are public ones, it would be nice to say it here and not later in the text (137). Also, as some patients have been chronic patients who were looked after more than 5 years, it would be nice to add it here.”

Response. Now, the suggested amendment is made accordingly (Please, see the study design, setting and period section).

Comment. “Lines 136-141. I would simplify the text, a suggestion: “Inclusion criteria: patients who were > 18 years, they had chronic diseases (hypertension, diabetic mellitus, and bronchial asthma) and they were on follow-up in the hospitals during the data collection period. Whereas unable to communicate via any channel, and admitted patients were excluded”.”

Response. We thank you so much for the suggestion and now, the amendment is made accordingly (Please, see the participants section).

Comment. “Lines 148-150 This information will be better located in the setting. On the other hand, are these clinics the only ones in the whole country, in the area? Please, clarify.”

Response. We accepted the comment and the suggested information is now located in the setting (Please, see the study design, setting and period section).

Comment. “Lines 150-152, it would be clearer if they describe the flow of patients in the clinics in the setting section in line 128-134.”

Response. Now, the information related with patients flow is describe in the setting (Please, see the study design, setting and period section).

Comment. “Lines 161-206. I found difficult to follow this section. I don’t know if I am wrong but I have understood that the authors created the test with sections of different surveys that have been administered in other studies. If this is the case, it would be nice to start with this clear information. I would suggest to add a table with the scores of each scale. It is difficult to interpret the results without a clear idea of how each variable was measured (169-206). Also finishing with the score of this final test.”

Response. We have modified the entire instrument and measurements section. In the meantime, we tried to address the raised issues in the instrument and measurements section (Please, see the uploaded Revised Manuscript with Track Changes and the Manuscript file)..

Comment. “Line 209. I would suggest to describe this section after sampling technique. I was confused reading line 162 about the pre-tested.”

Response. The data collection methods and quality assurance section is now located after sampling technique (Please, see the revised version of the manuscript).

Comment. “Line 292-296, As a suggestion: It would be interesting to know if patients who had several conditions are more prone to compliance the preventive measures.”

Response. We completely agree with your suggestion. However, their relationship has already been checked during the selection of candidate explanatory variables for final model which was not shown a statistically significant relationship. As a result, this variable was not included in the table (previously named table 5, currently modified as table 4) (please, see both the previous and the revised manuscript).

Comment. “The titles of the tables must be simplified. There is no need to describe the patients and the setting as both characteristics have been described in methods before.”

Response. Now, the titles of the tables are simplified as much as possible (Please, see the revised version of the manuscript). 

Comment. “In table 6, the average monthly income is difficult to understand for an international audience, please change it to USA dollars.”

Response. We accepted the comment and the average monthly income ETB is change to US Dollar in Table 1,4 and 5 (see the revised version of the manuscript).

Comment. “Lines 350-351 has already been said before.”

Response. Now, the stated objective is removed from the discussions section. 

Comment. “Lines 357-362, it would be needed to compare with similar countries to make the text more interesting for an international audience.”

Response. We accepted the comment and we have compared with similar countries (Please, see the discussions section of the revised manuscript).

Comment. “Lines 370-375. I would like a little bit of context about the COVID-19 health promotion in Ethiopia. Was there any national campaign to explain the protection measures? What kind of education both hospitals offered to their patients? May be patients were not protecting themselves because anyone explained them clearly how to do it. As mass media or religious leaders seem to have a role in the community, it would be nice to quote if any of the leaders in the community or in the country took a proactive role to share preventive measures.”

Response. We accepted the comment and the suggested information is now included (Please, see the introduction section and the discussions section of the revised manuscript). 

Comment. “Lines 424-434. I think this is a key result, clean water is a basic need and this study gives another motive to support the access to clean water in each home.”

Response. We thank you so much.

Comment. Line 440-451. If we are going to discuss the impact of having internet access. There should be a reference in the introduction regarding the use of internet in Ethiopia. Secondly, it would be helpful to understand how people in the community are using it, do they use more social media (Facebook, WhatsApp, Telegram, etc.) than other sources of information?

Response. We accepted the comment and the suggested information is now included in the introduction section of the revised manuscript. 

We thank you so much!

---

## [Decision Letter · Decision Letter 1]

30 Aug 2022

PONE-D-22-10853R1Compliance with COVID-19 preventive measures among chronic disease patients in Wolaita and Dawuro zones, Southern Ethiopia: A proportional odds modelPLOS ONE

Dear Dr. Gelgelu,

Thank you for submitting your manuscript to PLOS ONE. After careful consideration, we feel that it has merit but does not fully meet PLOS ONE’s publication criteria as it currently stands. Therefore, we invite you to submit a revised version of the manuscript that addresses the points raised during the review process.

ACADEMIC EDITOR:

Dear Author

Thank you very much for the effort in addressing al comments needed.

Please have a short review of some final minor comments before proceeding for final decision.

Best regards

We look forward to receiving your revised manuscript.

Kind regards,

Juan A López-Rodríguez

Academic Editor

PLOS ONE

Journal Requirements:

Reviewers' comments:

Reviewer's Responses to Questions

**Comments to the Author**

1. If the authors have adequately addressed your comments raised in a previous round of review and you feel that this manuscript is now acceptable for publication, you may indicate that here to bypass the “Comments to the Author” section, enter your conflict of interest statement in the “Confidential to Editor” section, and submit your "Accept" recommendation.

Reviewer #1: All comments have been addressed

Reviewer #2: All comments have been addressed

2. Is the manuscript technically sound, and do the data support the conclusions?

Reviewer #1: Yes

Reviewer #2: Yes

3. Has the statistical analysis been performed appropriately and rigorously? 

Reviewer #1: I Don't Know

Reviewer #2: Yes

4. Have the authors made all data underlying the findings in their manuscript fully available?

Reviewer #1: Yes

Reviewer #2: No

5. Is the manuscript presented in an intelligible fashion and written in standard English?

Reviewer #1: Yes

Reviewer #2: Yes

6. Review Comments to the Author

Reviewer #1: (No Response)

Reviewer #2: I think the authors have improved significantly the manuscript. The methods section is clearer now and the results are much easy to follow. In my opinion, the manuscript should be accepted, I only suggest a few small changes to make the document more attractive.

Line 72, a small misspelling, change for “severe”

Line 109, please write the number in USA dollars

Line 191, i would eliminate quoting that the server is Google drive as in some ethics committee would not agree about storing the data in Google drive.

Line 201-202, i would eliminate these lines to make the reading more easy. I would finish the sentence: “During an exit interview, responses to the 199 questions were validated, restricted, and labeled require because expressions such as 200 constraint, relevant, and requirements were added to the data”.

Line 232: please correct “ yes”

Line 280-281 can be eliminated, only add at the end of that paragraph that the analysis was performed by “ SPSS version 25.

Line 290, it is not necessary to repeat twice spss.

Line 451-456 seems more a message for changes in clinical practice. I would consider to include in the Implications for clinical practice and future research.

Line 468, i would appreciate if they could mention the countries

Line 493, if they could explain briefly what Ekub and Edir mean, it would be appreciated.

I would add a section of “ Implications for clinical practice and future research”. Line 592-597 should be added there.

I would suggest that the first paragraph of the discussion would be a summary of the study.

In table 2, I would suggest to describe categories as true, false, I don’t know. I would let overall knowledge level in a line and below, I would write both categories in one row each of them ( Good, Poor).

In table 4, I would write in the same box the number and the marginal percentage ( 315 (75.5)) so it is only one box with the information. As there were other variables with median and mean units, it would make the table more clean. A sign * can be written in each category so the reader can check the units

Conclusion, should be shorter and giving a very clear message. I appreciate the high methodology work the authors have made but it is too much information

7. PLOS authors have the option to publish the peer review history of their article (what does this mean?). If published, this will include your full peer review and any attached files.

Reviewer #1: No

Reviewer #2: No

---

## [Author Response · Author response to Decision Letter 1]

19 Sep 2022

PONE-D-22-10853R1

Compliance with COVID-19 preventive measures among chronic disease patients in Wolaita and Dawuro zones, Southern Ethiopia: A proportional odds model

We value and appreciate the comments and suggestions made by editor(s) and reviewers, which have greatly improved the manuscript's quality. Besides, we have prepared the following responses to address their comments and suggestions. 

Responses for editor(s)' comments:

 Journal Requirements:

Comment_1. Please review your reference list to ensure that it is complete and correct. If you have cited papers that have been retracted, please include the rationale for doing so in the manuscript text, or remove these references and replace them with relevant current references. Any changes to the reference list should be mentioned in the rebuttal letter that accompanies your revised manuscript. If you need to cite a retracted article, indicate the article’s retracted status in the References list and also include a citation and full reference for the retraction notice.

Response_1. After reviewing every cited work, now we have updated our reference lists. Besides, no retracted articles have been cited (Please, see the references section of the uploaded Revised Manuscript with Track). 

Responses for reviewers' comments:

 “Reviewer's Responses to Questions”

“Comments to the Author”:

“1. If the authors have adequately addressed your comments raised in a previous round of review and you feel that this manuscript is now acceptable for publication, you may indicate that here to bypass the “Comments to the Author” section, enter your conflict of interest statement in the “Confidential to Editor” section, and submit your "Accept" recommendation.”

Reviewer #1: All comments have been addressed

Reviewer #2: All comments have been addressed

Response: Thank you

“2. Is the manuscript technically sound, and do the data support the conclusions?”

“The manuscript must describe a technically sound piece of scientific research with data that supports the conclusions. Experiments must have been conducted rigorously, with appropriate controls, replication, and sample sizes. The conclusions must be drawn appropriately based on the data presented.”

Reviewer #1: Yes

Reviewer #2: Yes

Response. It is ok 

“3. Has the statistical analysis been performed appropriately and rigorously?”

Reviewer #1: I Don't Know

Reviewer #2: Yes

Response: We have performed the statistical analysis in accordance with PLOS ONE's SAMPL guidelines and it is shown in the methods and materials section.

“4. Have the authors made all data underlying the findings in their manuscript fully available?”

“The PLOS Data policy requires authors to make all data underlying the findings described in their manuscript fully available without restriction, with rare exception (please refer to the Data Availability Statement in the manuscript PDF file). The data should be provided as part of the manuscript or its supporting information, or deposited to a public repository. For example, in addition to summary statistics, the data points behind means, medians and variance measures should be available. If there are restrictions on publicly sharing data—e.g. participant privacy or use of data from a third party—those must be specified.”

Reviewer #1: Yes

Reviewer #2: No

Response: In addition to the previous dataset, we have currently uploaded English version an ODK Excel form as Supporting Information files (Please, see previously uploaded minimal dataset and currently uploaded English version an ODK Excel form).

“5. Is the manuscript presented in an intelligible fashion and written in standard English?”

“PLOS ONE does not copyedit accepted manuscripts, so the language in submitted articles must be clear, correct, and unambiguous. Any typographical or grammatical errors should be corrected at revision, so please note any specific errors here.”

Reviewer #1: Yes

Reviewer #2: Yes

Response: Thank you

 “6. Review Comments to the Author”

Comment. “Please use the space provided to explain your answers to the questions above. You may also include additional comments for the author, including concerns about dual publication, research ethics, or publication ethics. (Please upload your review as an attachment if it exceeds 20,000 characters)”

Reviewer #1: (No Response)

Response. Thank you very much for your previous valuable contributions. 

 Reviewer #2: I think the authors have improved significantly the manuscript. The methods section is clearer now and the results are much easy to follow. In my opinion, the manuscript should be accepted, I only suggest a few small changes to make the document more attractive.

Response: We thank you so much for your acknowledgement and valuable suggestions. In the current revised manuscript, we have addressed your suggestions (Please, see the following responses for the respective comments). 

Comment. “Line 72, a small misspelling, change for “severe””

Response. It is corrected (Please, see the uploaded Revised Manuscript with Track Changes; line 57). 

Comment. “Line 109, please write the number in USA dollars”

Response. Now, it is done (Please, see the uploaded Revised Manuscript with Track Changes; line 96).

Comment. “Line 191, i would eliminate quoting that the server is Google drive as in some ethics committee would not agree about storing the data in Google drive.”

Response. Now, it is done (Please, see the uploaded Revised Manuscript with Track Changes; line 169-170).

Comment. “Line 201-202, i would eliminate these lines to make the reading more easy. I would finish the sentence: “During an exit interview, responses to the 199 questions were validated, restricted, and labeled require because expressions such as 200 constraint, relevant, and requirements were added to the data”.”

Response. The comment is accepted and the suggested modification is made (Please, see line 178-180 of the Revised Manuscript with Track Changes). In addition, as Supporting Information file, English version an ODK Excel form is also uploaded.

Comment. “Line 232: please correct “ yes”.”

Response. While renaming the categories to address other comment, in the meantime this issue is also corrected (Please, see the uploaded Revised Manuscript with Track Changes; line 201).

Comment. “Line 280-281 can be eliminated, only add at the end of that paragraph that the analysis was performed by “ SPSS version 25.”

Response. Now, the suggested amendment is made accordingly (Please, see the Revised Manuscript with Track Changes; line 226-230).

Comment. “Line 290, it is not necessary to repeat twice spss.”

Response. Now, it is deleted (Please, see the Revised Manuscript with Track Changes; line 236).

Comment. “Line 451-456 seems more a message for changes in clinical practice. I would consider to include in the Implications for clinical practice and future research.”

Response. Currently, we have added a new section implication for clinical practice and future research in the manuscript. Therefore, the suggested line is now incorporated into this section (Please, see the Revised Manuscript with Track Changes; line 494-502). 

Comment. “Line 468, i would appreciate if they could mention the countries”

Response. The countries are now mentioned (Please, see line 390-391 of the Revised Manuscript with Track Changes).

Comment. “Line 493, if they could explain briefly what Ekub and Edir mean, it would be appreciated.”

Response. Now, Ekub and Edir are explained (Please, see the Revised Manuscript with Track Changes; line 417-419).

Comment. “I would add a section of “ Implications for clinical practice and future research”. Line 592-597 should be added there.”

Response. Now, the suggested line is incorporated into the implications for clinical practice and future research section (Please, sees the Revised Manuscript with Track Changes; line 494-502).

Comment. “I would suggest that the first paragraph of the discussion would be a summary of the study.”

Response. The paragraph that used to summarize the predictors of compliance level in the discussion section line 426-432 of the Revised Manuscript with Track Changes is now moved to the first paragraph of the discussion. In the meantime, the suggested information is addressed accordingly (Please, see the first paragraph of the discussion section of the Revised Manuscript with Track Changes; line 368-375).

Comment. “In table 2, I would suggest to describe categories as true, false, I don’t know. I would let overall knowledge level in a line and below, I would write both categories in one row each of them ( Good, Poor).”

Response. Currently, Yes, No, I don’t know categories are described as True, False, I don’t know, respectively (Please, see table 2 of the Revised Manuscript with Track Changes). Besides, to make the description consistent with table 2 the statement that found in the line 201 is also modified.

Comment. “In table 4, I would write in the same box the number and the marginal percentage ( 315 (75.5)) so it is only one box with the information. As there were other variables with median and mean units, it would make the table more clean. A sign * can be written in each category so the reader can check the units”

Response. Now, table 4 is modified accordingly (Please, see table 4 of the Revised Manuscript with Track Changes).

Comment. “Conclusion, should be shorter and giving a very clear message. I appreciate the high methodology work the authors have made but it is too much information”

Response. We accepted the comment and the conclusions section is modified accordingly (Please, see the conclusions section of the Revised Manuscript with Track Changes; line 521-535).

 “I like the article. Those are some suggestions. The thing that bothers me is the n number. Size sampled calculate to be 423, the get 419, and only work with 417. I don´t see clearly how you get to 417.”

Response. Thank you very much for the acknowledgement. Regarding your concern, 419 out of 423 (99%) participants were willing to take part in the study, while 4 were not. Accordingly, data related to socio-demographic and clinical characteristics, knowledge of the mode of transmission of COVID-19 and access to water and sanitation status were collected from 419 study participants. In addition to the previously obtained data, participants who had information of the recommended COVID-19 preventive measures were asked to provide data related to the recommended action. Consequently, 2 participants had no information about preventive measures that were recommended to adhere to), whereas 417 had. As a result, data related to their perceived health beliefs about the advised action, their attitude towards the advised action, and their level of compliance with the recommended preventive measures were collected from 417 study participants. Besides, to address the concern, a paragraph that describes the procedure is now added at the beginning of the results section along with its flow diagram (Please, see separately uploaded flow diagram (Fig 2) and the Revised Manuscript with Track Changes; line 277-286). 

Comment. “Line 57: “sever form” , probably should be “severe” “

Response. Now, it is corrected (Please, see the uploaded Revised Manuscript with Track Changes; line 57).

Comment. “Line 58: Space after the dot “(10).Studies””

Response. Now, it is corrected (Please, see the uploaded Revised Manuscript with Track Changes; line 57-58).

Comment. “Line 60 “while a chronic respiratory disease was also indicated in the list”. This sentence is a bit alone and could be fused with the previous one e.g. “most frequent [or some of the most frequent] chronic conditions were hypertension, diabetes, and chronic respiratory diseases … “”

Response. The comment is now addressed as suggested (Please, see the uploaded Revised Manuscript with Track Changes; line 59-61).

Comment. “Line 67 It was also proven effective → It also has been proven effective”

Response. Now, it is changed to the suggested statement (Please, see currently Revised Manuscript with Track Changes; line 69). 

Comment. “Line 80 (and others along the text): the apostrophes denoting possession are straight “experts`..." should be experts’...”. Also used that way in other parts of the text. 401 participants` 429”

Response. We thank you very much for the comment and the issue is now fixed in line 81, 82, 101, 136, 183, 207, 208, 218, 222, 257, 271, 310, 322, 394, 422,436, 452, 464, 476, 488, Table 2 (I don’t know) of the Revised Manuscript with Track Changes.

Comment. “Line 81: “community” is singular, therefore “community was” (not were)”

Response. Really thanks for the comment. It is corrected now (Please, see currently Revised Manuscript with Track Changes; line 83).

Comment. “Line 94 (and other parts of the text) : in the text the use of symbols like < , > , = should be avoided, but can be used in tables. 144, 278”

Response. The issue is now fixed in line 95-96, 147, 194, 195, 205, 245, 258-259, 260, 267, 295, 333, 334 of the Revised Manuscript with Track Changes.

Comment. “Line 111: “so as the finding can help to highlight WASH potential gaps in the study area.” → maybe it would be better to shorten it to better understand it to “so as the finding can help to highlight WASH potential gaps in the study area.” Also, I don’t understand why WASH is in capital letters.”

Response. We value and respect your comment. WASH “is an acronym that stands for the interrelated areas of “Water, Sanitation and Hygiene”, and it is used widely by non-governmental organizations and aid agencies in developing countries.” As a result, in our study it was used as an acronym to indicate the general gaps that related with access to water and sanitation. Now, the statement is rewritten to facilitate the readers understanding and to address the comment (Please, see the Revised Manuscript with Track Changes; line 113-114).

Comment. “144

“who were older than 18 years” or “18 years or older”? 

they had chronic diseases → who had chronic diseases → or even “who had hypertension, diabet…. and..””

Response. Now, the comments are addressed (Please, see the Revised Manuscript with Track Changes; line 147-148).

Comment. “146 Whereas unable to communicate → Whereas patients unable..”

Response. Now, the comment is addressed (Please, see the Revised Manuscript with Track Changes; line 149-150).

Comment. “148 Sample size. It surprised me that after calculating a required sample of 423, you got 419 and later ended with 417 . It was a good idea to add a 10% non response rate. Have you considered adding a patient flow chart?”

Response. Of course the determined sample size was 423. To address this comment, a paragraph that describes the procedure is now added at the beginning of the results section along with its flow diagram (Please, see separately uploaded flow diagram (Fig 2) and the Revised Manuscript with Track Changes; line 277-286).

Comment. “189 … by adding each score out of 44 → by adding each score up to 44?? “

Response. Now, the comment is addressed (Please, see the Revised Manuscript with Track Changes; line 192-193).

Comment. “197 Ye/No → Yes/no?”

Response. To address the other comment, the Yes, No, I don’t know categories are now replaced with True, False, I don’t know, respectively. Therefore, this comment is also addressed in the meantime (Please, see the Revised Manuscript with Track Changes; line 201). 

Comment. “206 … a higher score (2 point) → the highest score??”

Response. Now, the comment is addressed (Please, see the Revised Manuscript with Track Changes; line 210).

Comment. “214*215 This questionnaire was attested for content validity by health education and public health experts. →bibliography supporting that quote?”

Response. Now, the affiliation of the professionals is included in the statement to better describe them (Please, see the Revised Manuscript with Track Changes; line 219-220).

Results

Comment. “271 About the n, how come you calculated a study size of 423 and only got 419 patients. What happened with the other 4?”

Response. As you indicated, the determined sample size was 423. However, during the interview session 423 eligible study participants were asked for their consent to participate in the study. In the meantime, 419 out of 423 (99%) participants were willing to take part in the study, while 4 were not. Besides, a paragraph that describes the procedure is now added at the beginning of the results section along with its flow diagram (Please, see separately uploaded flow diagram (Fig 2) and the Revised Manuscript with Track Changes; line 277-286).

Comment. “273 majorities of them → majority of them”

Response. The comment is currently addressed (Please, see the Revised Manuscript with Track Changes; line 289-290).

Comment. “274…, and employed 282 → and were employed . Here employed is not used as verb but as an adjective.”

Response. Now, the comment is addressed (Please, see the Revised Manuscript with Track Changes; line 291).

Comment. “274-276 In this study, television was mentioned as a major 362 (86.8%), while the website was mentioned as the least 96 (23.0%) source of information about COVID-19 preventive measures.

Changing the order of one line improves its understanding → In this study, television was mentioned as a 275 major 362 (86.8%)source of 276 information about COVID-19 preventive measures , while the website was mentioned as the least 96 (23.0%).”

Response. The comment is accepted and addressed accordingly (Please, see the Revised Manuscript with Track Changes; line 292-294).

Comment. “279 It surprised me to see all the tables with the year 2021 on it. Not used to see that.

Table 1 When you say “Age in years” → “Age (years)”

In the table it is appropriate to use symbols >, < and so on”

Response. We respect your comment. However, this was done while we tried to fulfil the notion that indicates a good table-title should answer a What, When, Where, How classified question. Now, it is revised (Please, see all tables of the Revised Manuscript with Track Changes).

Comment. “295 TAble 1 and table 2: n=419 but when you get to line 295 you say “417 had information about covid-19 preventive measurements”. How do you get to this n? Much later in the line 352 you say “almost all (417 out of 419) participants had information about covid.”

Response. In the study, 419 out of 423 (99%) participants were willing to take part, while 4 were not. Accordingly, characteristics such as socio-demographic and clinical, and knowledge towards the mode of transmission of COVID-19 were calculated, using data from 419 study participants (this makes TAble 1 and table 2: n=419). In the meantime, almost all (417 out of 419, 99.5%) participants did meet the established precondition (they had information about COVID-19 preventive measures that were recommended to adhere to), while 2 participants did not. Besides, a paragraph that describes the procedure is now added at the beginning of the results section along with its flow diagram (Please, see separately uploaded flow diagram (Fig 2) and the Revised Manuscript with Track Changes; line 277-286).

Comment. “297 More than a half; (59.0%) of participants → More than half of the participants (59,0%)

The semicolon (;) should be erased” 

Response. The comment is addressed accordingly (see Revised Manuscript with Track Changes; line 316-317).

Comment. “298 (51,6%) → erase parenthesis”

Response. The parenthesis is erased as commented (see Revised Manuscript with Track Changes; line 318).

Comment. “346 In this study, the determined compliance level and the identified predictors will have a valuable contribution to the current level of knowledge.

I believe this sentence should be rewritten for better understanding and also fused with the next one →

The determination of the level of compliance with the covid19 preventive measures and its predictors in patients with chronic illnesses are a valuable contribution that will help….”

Response. The sentence is rewritten as suggested (see Revised Manuscript with Track Changes; line 495-498).

Comment. “352 417/419 → 417 out of 419”

Response. It is corrected (see Revised Manuscript with Track Changes; line 382).

Comment.“354 majorities of the particip→the majority of the participants 360 ….”

Response. It is corrected (see Revised Manuscript with Track Changes; line 384).

Comment. “364 Out of the blue → used more frequently in an informal way → might be better to use “Unexpectedly” or similar. Your pick”

Response. It is corrected (see Revised Manuscript with Track Changes; line 396).

Comment. “365- 366 responded, blood transfusion as a means → responded blood transfusion was a …

responde that sexual intercourse as another → was another.. “

Response. It is corrected (see Revised Manuscript with Track Changes; line 397-398).

Comment. “366 Findings from Ethiopia → Other studies from Ethiopia ….”

Response. It is corrected (see Revised Manuscript with Track Changes; line 398-399).

Comment. “367 Findings from Ethiopia and elsewhere showed that (12.38% and 7.7%) of participants responded to blood transfusion, while (9% and 7.4%) of participants, responded to sexual intercourse as a means of COVID-19 transmission → I would rephrase this sentence to improve its understanding

May be something like → Other studies from Ethiopia and elsewhere showed that 12.38% and 7.7% of participants responded blood transfusion was a mean of COVID-19 transmission while 9% and 7.4% of participants answered sexual intercourse was another mean. [I’m sure you can even make a better sentence]

For sure, take away the parenthesis and don’t say “as a means” , “a “ is singular while “means” is plural.”

Response. It is corrected (see Revised Manuscript with Track Changes; line 398-402).

Comment. “372-373 and after →and after it the government has deployed →has or had?”

Response. Now, it is corrected (see Revised Manuscript with Track Changes; line 406).

Comment. “409

a) it needs → he/she needs

b) to believe that personally susceptible → to believe he/she is personally

c) susceptible to a health → susceptible to such health ??”

Response. Now, it is corrected (see Revised Manuscript with Track Changes; line 444-445).

Comment. “415 Implies the more → Which implies?”

Response. Now, it is corrected (see Revised Manuscript with Track Changes; line 450).

Comment. “429 COVID-19, in turn it helps → COVID-19, to probably help the”

Response. Now, it is corrected (see Revised Manuscript with Track Changes; line 464-465).

Comment. “432 the pieces of evidence →the evidence revealed”

Response. It is corrected (see Revised Manuscript with Track Changes; line 467).

Comment. “433 in a home, in turn, would help → in a home, which would help”

Response. It is corrected (see Revised Manuscript with Track Changes; line 468).

Comment. “484 strategies by considering → strategies considering”

Response. Since this paragraph is incorporated in the implications for clinical practice and future research section, this comment is addressed in the meantime (see Revised Manuscript with Track Changes; line 494-502). 

Comment. “488 in the study setting → in any setting.”

Response. It is corrected (see Revised Manuscript with Track Changes; line 502).

Comment. “FIG 1 It’s repeated, apostrophe incorrect “participants`". Has the year on it: 2021. I’m not used to see it on a table nor a figure.”

Response. Now, it is corrected (Please, see separately uploaded Fig 1).

Comment. “Bias

Only people who attended the follow up clinic sesgo de gente preocupada porsu salud” 

Response. The comment is accepted, and it is noted in the study’s limitations (Please, see Revised Manuscript with Track Changes; line 513-515).

We thank you very much!

---

## [Decision Letter · Decision Letter 2]

10 Oct 2022

Compliance with COVID-19 preventive measures among chronic disease patients in Wolaita and Dawuro zones, Southern Ethiopia: A proportional odds model

PONE-D-22-10853R2

Dear Dr. Gelgelu,

We’re pleased to inform you that your manuscript has been judged scientifically suitable for publication and will be formally accepted for publication once it meets all outstanding technical requirements.

Kind regards,

Juan A López-Rodríguez

Academic Editor

PLOS ONE

Additional Editor Comments (optional):

Reviewers' comments:

Reviewer's Responses to Questions

**Comments to the Author**

1. If the authors have adequately addressed your comments raised in a previous round of review and you feel that this manuscript is now acceptable for publication, you may indicate that here to bypass the “Comments to the Author” section, enter your conflict of interest statement in the “Confidential to Editor” section, and submit your "Accept" recommendation.

Reviewer #1: All comments have been addressed

Reviewer #2: All comments have been addressed

2. Is the manuscript technically sound, and do the data support the conclusions?

Reviewer #1: Yes

Reviewer #2: Yes

3. Has the statistical analysis been performed appropriately and rigorously? 

Reviewer #1: I Don't Know

Reviewer #2: Yes

4. Have the authors made all data underlying the findings in their manuscript fully available?

Reviewer #1: Yes

Reviewer #2: Yes

5. Is the manuscript presented in an intelligible fashion and written in standard English?

Reviewer #1: Yes

Reviewer #2: Yes

6. Review Comments to the Author

Reviewer #1: Those are minor revisions to improve the paper. Just some typing errors. I like the work you have done.

Reviewer #2: The authors have answered all the suggestions in the last review, the paper should be published if the editor agreed.

7. PLOS authors have the option to publish the peer review history of their article (what does this mean?). If published, this will include your full peer review and any attached files.

Reviewer #1: No

Reviewer #2: No

---

## [Editor Report · Acceptance letter]

17 Oct 2022

PONE-D-22-10853R2 

Compliance with COVID-19 preventive measures among chronic disease patients in Wolaita and Dawuro zones, Southern Ethiopia: A proportional odds model 

Dear Dr. Gelgelu:

I'm pleased to inform you that your manuscript has been deemed suitable for publication in PLOS ONE. Congratulations! Your manuscript is now with our production department. 

Kind regards, 

on behalf of

Dr. Juan A López-Rodríguez 

Academic Editor

PLOS ONE